# Long-Term Reduction of Bacterial Adhesion on Polyurethane by an Ultra-Thin Surface Modifier

**DOI:** 10.3390/biomedicines10050979

**Published:** 2022-04-23

**Authors:** Brian De La Franier, Dalal Asker, Benjamin Hatton, Michael Thompson

**Affiliations:** 1Department of Chemistry, University of Toronto, 80 St. George Street, Toronto, ON M5S 3H6, Canada; brian.delafranier@mail.utoronto.ca; 2Department of Materials Science, University of Toronto, 184 College Street, Toronto, ON M5S 3E4, Canada; dalal.asker@utoronto.ca or dasker10@gmail.com (D.A.); benjamin.hatton@utoronto.ca (B.H.); 3Food Science and Technology Department, Faculty of Agriculture, Alexandria University, Alexandria 21545, Egypt

**Keywords:** microbes, fouling, anti-fouling, monoethylene glycol, polyurethane biofilms, long term

## Abstract

Indwelling urinary catheters are employed widely to relieve urinary retention in patients. A common side effect of the use of these catheters is the formation of urinary tract infections (UTIs), which can lead not only to severe medical complications, but even to death. A number of approaches have been used to attempt reduction in the rate of UTI development in catheterized patients, which include the application of antibiotics and modification of the device surface by coatings. Many of these coatings have not seen use on catheters in medical settings due to either the high cost of their implementation, their long-term stability, or their safety. In previous work, it has been established that the simple, stable, and easily applicable sterilization surface coating 2-(3-trichlorosilylpropyloxy)-ethyl hydroxide (MEG-OH) can be applied to polyurethane plastic, where it greatly reduces microbial fouling from a variety of species for a 1-day time period. In the present work, we establish that this coating is able to remain stable and provide a similarly large reduction in fouling against *Escherichia coli* and *Staphylococcus aureus* for time periods in an excess of 30 days. This non-specific coating functioned against both Gram-positive and Gram-negative bacteria, providing a log 1.1 to log 1.9 reduction, depending on the species and day. This stability and continued efficacy greatly suggest that MEG-OH may be capable of providing a solution to the UTI issue which occurs with urinary catheters.

## 1. Introduction

In order to relieve urinary retention in hospitals and care home settings, indwelling urinary catheters are used, with over 100 million being sold worldwide annually [1]. Despite their widespread use, these devices are not without serious complications, with the development of nosocomial urinary tract infections (UTIs), generally known as catheter-associated UTIs (CAUTIs) being the most serious among them. Roughly 80% of catheterized patients will develop these infections, which can lead to not only severe medical complications and death, but also to significant economic cost in terms of treatment of the resulting infections [2,3,4,5]. There are several species that can cause CAUTIs, with *Escherichia coli*, *Proteus mirabilis*, *Pseudomonas aeruginosa*, *Candida albicans*, and *Staphylococcus aureus* being among the most common [1].

Several approaches to resolving this issue of CAUTIs have been proposed, with various surface coatings being the most common. One of these approaches is the use of polyethylene-glycol chains (PEG) [6,7,8,9,10,11,12,13], which reduce the fouling observed from bacteria by between 80 and 94%, depending on the species studied. However, these coatings have the downside of reduced long-term stability and eventual breakdown, potentially causing cytotoxic materials to enter the body [14,15,16,17,18]. Additionally, slippery liquid-infused porous surfaces (SLIPS) based coatings have also been studied [19,20,21,22], where the amount of fouling reduction depends highly on the surface roughness and oil used and shows anywhere from a 50 to 99% reduction. Despite their reasonable performance against fouling, these coatings are typically difficult to apply, and cause leeching of the infused oil [23], which has limited their use in the medical setting.

Surface bound enzymes have also been explored [24,25,26] and show high reduction of 98–99.9%, but the denaturation of proteins and their inability to be sterilized by traditional methods have also limited their use [27,28]. In addition, surface-bound nanoparticles have been extensively studied [29,30,31,32], showing a 70–90% reduction in bacterial fouling depending on the pH and particle used, but their cytotoxicity is of great concern for their use in implanted medical devices [33,34,35]. A more detailed review of these coatings and their effects has been described previously [36].

A surface coating that is low cost to produce, is stable for long periods of time, does not leach potentially toxic compounds, and can be sterilized by traditional methods is needed to solve the issue of CAUTIs. In our research, we have developed and investigated a mono-ethylene glycol-based surface coating, which uses a silane linkage, 2-(3-trichlorosilylpropyloxy)-ethyl hydroxide (MEG-OH) to modify a variety of materials (Figure 1) [37,38]. This coating has proven effective at reducing fouling from blood cells and serum proteins [39,40,41]. This coating functions by allowing a semi-crystalline hydration layer to form at the surface of the coated material, which prevents fouling [42,43].

In a recently published work, this coating was shown to reduce microbial fouling by *E. coli, S. aureus, P. aeruginosa,* and *C. albicans* for a 1-day time period when attached to polyurethane tubing [35]. In addition, it was found to retain the capability to reduce fouling after autoclaving, after long-term storage, and across a 3-day time period. Additionally, this coating is low cost to produce and easy to apply to a variety of materials as shown in our previous work. The simple structure of MEG-OH suggests that it will not cause toxic compounds to release from the surface, though cytotoxicity studies have yet to be performed. This gives MEG-OH a very significant advantage over the other discussed coatings as a potentially viable medical coating.

In order to be effective against the formation of CAUTIs in patients, MEG-OH coatings would have to be stable for several weeks on surfaces, and still provide anti-microbial fouling across that time period. In order to test that in this work, a CDC bioreactor was used to incubate modified and control polyurethane samples in a low concentration lysogenic broth medium that had been infected with either Gram-negative *E. coli*, or Gram-positive *S. aureus* bacteria, allowing for samples to be tested for bacterial fouling for a time period greater than 1 month. It is hypothesized that MEG-OH will provide a similar reduction in fouling across this time period, as was observed in those previous 1-day trials.

## 2. Materials and Methods

### 2.1. Materials

MEG-TFA was synthesized according to previously published methods [37]. Heptane was purchased from Sigma-Aldrich (St. Louis, MO, USA). Polyurethane tubing was purchased from McMaster Carr (Aurora, OH, USA). Ethanol was obtained from Caledon Laboratory Chemicals (Georgetown, ON, Canada). Broths bases, agar, and fetal bovine serum (FBS) were purchased from University of Toronto MedStore (Toronto, ON, Canada). Glutaraldehyde (GDA) and Tween-20 (T20) were purchased from Sigma-Aldrich (Mississauga, ON, Canada). Staining was performed with SYTOX green (Life Technologies, Carlsbad, CA, USA). Adhesion experiments were carried out with *Escherichia coli* GFP (ATCC 25922GFP) and *Staphylococcus aureus* KR3.

### 2.2. Polymer Surface Modification

Polyurethane tubing (1/16″ inner diameter, 1/8″ outer) was cut into 2.5 cm sections. These sections were then rinsed with 1% SDS, followed by 5 min of sonication in 1% SDS, and a final rinse of 1% SDS. They were then rinsed with 95% ethanol, followed by sonication in 95% ethanol, and a final rinse in 95% ethanol. After this, the samples were dried under a stream of air. Control samples were set aside at this point, while experimental samples were given 10 min of plasma cleaning under atmosphere twice, rotating the samples between cleans. The experimental samples were then stored in a humidity chamber held at 80% relative humidity overnight.

In pre-silanized glass scintillation vials, the samples were completely submerged in a MEG-TFA solution (2:1000 MEG-TFA:heptane) and placed on a rotator for 90 min under N_2_ atmosphere. After this, the solution was removed from each sample, followed by rinsing each with heptane, and sonicating each in heptane for 5 min. After this, each sample was again rinsed with heptane, then rinsed with 95% ethanol. The samples were then left rotating overnight in a solution of 50% ethanol in water to deprotect the surface adlayer. The MEG-OH-coated samples were rinsed with 95% ethanol, then air dried and stored until use.

### 2.3. Bacterial Incubation and Visualization

To verify the integrity of the surface coating, the samples were first tested for 24 h. To do so, a polyurethane control and MEG-OH-coated tubing were placed in a 6-well plate (2 tubes of the same condition in each well). The tubes were pre-fouled with fetal bovine serum (150 μL/tube) for 1 h to simulate natural protein fouling. Standard culture conditions were used to prepare all bacteria for adhesion. Pre-cultures of the tested strains were prepared by obtaining a single colony of freshly grown cells from a lysogenic broth (LB) agar media plate incubated overnight at 37 °C. This colony was inoculated into 5 mL of LB and incubated overnight at 37 °C under constant agitation. Then, 5 mL of bacterial solution was added to each well (1% *S. aureus* in LB, or 1% *E. coli* in LB). The well plate was incubated at 37 °C for 24 h without shaking. Each tube was then removed from its well and rinsed with PBS solution (3 × 10 mL), followed by pipetting 2 mL of PBS through the tube, and stored in a sterile 24-well plate.

A 1 cm section of each tube was cut from the center of the sample and placed in 10 mL of PBS solution, which was then sonicated for 5 min to release any bound organisms. This solution was then serially diluted to 100X and 1000X. For each concentration, including the original 10 mL solution, 3 spots of 20 μL each were spotted onto an LB agar plate. The plates were incubated for 18 h at 37 °C, and then the grown colonies were counted.

The remaining sections of tubing were then submerged in 1% GDA saline solution (1 mL per plate) for 30 min. After fixing the bacteria with GDA, the plates were then removed from the GDA solution and placed in a 0.05% Tween-20 in 0.9% NaCl for 10 min, then stained in the dark for 30 min with 50 µL Sytox Green in PBS buffer. The Sytox green solution was diluted from each tube with 0.9% NaCl solution (2 mL per plate). The tubes were then imaged by fluorescence microscopy (Olympus BX63, Tokyo, Japan) using 4X, 10X, and 20X dry objectives with a GFP filter (λ_ex_/λ_em_ 395/470 nm). Image analysis and filtering was performed with the Olympus cellSens imaging software (3.2, Olympus, Tokyo, Japan).

### 2.4. Long Term Incubation

For long-term anti-fouling studies, control and MEG-OH samples were first pre-fouled with serum as previously described. The samples were then placed in the standard CDC bioreactor sample holders by bending the tubes and pushing them into the cylindrical openings. The elastic rebound of the tubes was sufficient to hold them in place within the bioreactor. The reactor was then filled with 400 mL of bacteria solution (1% *S. aureus* in LB, or 1% *E. coli* in LB) and allowed to incubate overnight at room temperature.

Following this, a dilute medium was flowed into the reactor (2 g/L LB, 2 mL/min), and the flow was continuous for the entirety of the experiment. The solution was not stirred in order to better simulate the low-flow conditions of a catheter and bag. Two samples of each condition were taken from the bioreactor and processed using the previously described methods at various time points, up to 35 days. Statistical analysis of the agar plate counts using a two variable *t-*test (*n* = 6, df = 10) was performed for each sampled day.

## 3. Results

### 3.1. Reduction of E. coli Fouling

In order to study the long-term fouling of *E. coli* (Gram-negative bacteria) versus both control and MEG-OH-coated polyurethane, samples were taken from the bioreactor 24 h after incubation with a full concentration LB medium (static culture), then at days 14, 28, and 35 under the flow of 2 g/L LB medium. For these samples, a 1 cm section was cut from the middle of the sample and used for CFU counting (Figure 2), and the remainder of the sample was stained with Sytox green fluorescent stain and visualized at 20× magnification with fluorescence microscopy (Figure 3).

CFU counts and fluorescence images both show a reduction in *E. coli* fouling as a result of MEG-OH coatings, compared to control surfaces for the entire time period studied. After 1 day in a full concentration medium, much of the control PUR surface is covered in loosely connected bacteria and has a CFU count of 5.8 × 10^5^ CFU/cm^2^. Meanwhile the MEG-OH coated PUR surface shows only a handful of bacteria on the surface with no connections between them and has a CFU count of 0.4 × 10^5^ CFU/cm^2^, which represents a log 1.2 reduction in fouling as a result of MEG-OH, with a 98% confidence interval.

After 2 weeks in a diluted medium under flow conditions, the number of bacterial cells on the control surface appears to be reduced to some extent, and a CFU count of 2.1 × 10^5^ CFU/cm^2^ colonies. Still, the MEG-OH-coated surface shows an even greater reduction in bacteria, with very few visible at all across the surface, and a much lower CFU count of 0.3 × 10^5^ CFU/cm^2^, which is a similar log 1.1 reduction in fouling, with a 99.9% confidence interval.

Four weeks following the change to flowing diluted medium, the number of bacteria on the control surface appears to have increased compared to the 2-week samples, and interestingly, the individual bacteria appear to each be smaller. This is confirmed with a CFU count of 3.1 × 10^5^ CFU/cm^2^. The MEG-OH samples still show a large visual reduction in fouling, and a much lower CFU count of 0.3 × 10^5^ CFU/cm^2^, again, a log 1.1 reduction in fouling, with a 99% confidence interval.

Finally, after 5 weeks in the lower concentration medium, a thicker biofilm of bacteria is visible on the control samples, with brightly visible bacteria suspended throughout and a CFU count of 29.6 × 10^5^ CFU/cm^2^. This biofilm is non-existent on the MEG-OH samples, which do show a larger number of bacteria compared to previous weeks, it is still highly reduced compared to the control samples with a CFU count of 1.6 × 10^5^ CFU/cm^2^, a log 1.3 reduction, with a 99.8% confidence interval.

### 3.2. Reduction of S. aureus Fouling

As with the *E. coli* month-long study, an identical study was performed with *S. aureus* (Gram-positive bacteria) as the instigating bacteria. As with the *E. coli* samples, CFU counts were prepared from 1 cm of tubing (Figure 4), and the remainder was used for fluorescence imaging (Figure 5).

The day 1 counts showed a reduction from 16.6 × 10^4^ CFU/cm^2^ for the control samples, to 0.9 × 10^4^ CFU/cm^2^ for the MEG-OH modified samples, which represents a log 1.2 reduction in fouling with a confidence interval of 95%, which agrees well with the fluorescence images of these samples.

As with *E. coli,* there was an initial drop in colony counts at the 2 week time period, or 12 days in this case, following the reduction in media concentration. The counts decreased to 1.1 × 10^4^ CFU/cm^2^ for the control samples, and 0.2 × 10^4^ CFU/cm^2^ for the MEG-OH samples, which is a log 0.7 reduction in observed fouling with a confidence interval of 95%. Despite the overall lower counts, the fluorescence images show some biofilm formation on the control samples, which was not observed on day 1, despite the lower CFU counts.

For the following time periods, the levels of fouling for control samples increased at each timepoint to 2.3 × 10^4^ CFU/cm^2^ at day 19, and 3.3 × 10^4^ CFU/cm^2^ at day 33. However, for MEG-OH, the levels of fouling decreased across each timepoint to 0.1 × 10^4^ CFU/cm^2^ at day 19, and 0.5 × 10^3^ CFU/cm^2^ at day 33. This means that the reduction in anti-fouling provided by MEG-OH increased to log 1.3 at day 19, and log 1.9 at day 33, with 90% and 99% confidence intervals, respectively. These findings agree with the fluorescence images, which show very few bacteria on the MEG-OH samples, while control samples show areas of large colonies, and biofilm formation.

## 4. Discussion

For both experiments, beginning with either *E. coli* or *S*. *aureus* as the seed bacteria, MEG-OH showed a greater than log 1 reduction in fouling with a significant confidence interval across nearly every time point sampled for the more than 30 days that the experiments were performed. Additionally, both experiments showed predominantly individual bacteria across much of the control surfaces after 24 h of exposure, with *E. coli* showing far more bacteria in both the fluorescence images and the CFU counts. This matches well with the 24 h experiments published previously regarding MEG-OH anti-microbial fouling [36].

After the first 2 weeks in diluted media, both *E. coli-* and *S. aureus*-exposed tubing showed reductions in the colony counts for control and MEG-OH tubing. This is most likely due to the reduction in available food in the lower concentration media causing fewer bacteria to be able to survive. In the case of *S. aureus,* the reduction in fouling being reduced to 82% at this time point is likely due to the much lower amount of bacteria fouling the control surface as opposed to a reduction in the efficacy of MEG-OH, especially given that the percent reduction in colony counts returned to greater than 90% in subsequent weeks.

Over time, the number of bacteria fouling the control surfaces for both species began to recover, potentially due to biofilm formation. This is evidenced by the reduction in bacteria size visible for *E. coli*-coated control tubing at days 28 and 35, as well as additional biofilm being visible at later time points. Additional experiments would be needed to determine if this is the case, but that is beyond the scope of this investigation.

Instead, what is important is the reduction in surface-bound bacteria and biofilm formation observed on PUR tubing as a result of the MEG-OH coating. The greater than log 1.3 reduction in observed counted and observed colonies for both *E. coli* and *S. aureus* seeded bacteria after more than one month compares very favorably to PEG coatings [14], which were found to reduce *E. coli* biofouling approximately log 1 for a similar time period, while also showing signs of coating degradation.

These results also compare very favorably to SLIPS-coated surfaces, which typically release the majority of their compounds within the first few hours–days [19] and show biofouling along areas where the oil has defused [20]. As MEG-OH does not require any compound to elute from the surface to be effective and the monolayer itself is easy to apply and uniform, this is not an issue for this coating. Instead, the effective anti-fouling ability of the MEG-OH coating was confirmed to last for more than one month, which suggests as well that the coating is stable during this time.

This bodes well for the use of MEG-OH as a coating for urinary catheters to reduce bacterial fouling and potentially CAUTIs, as those catheters are typically not implanted for longer than one month. Since MEG-OH is able to be sterilized prior to use [36] and displays stable anti-fouling against both Gram-positive and Gram-negative seeded bacteria for longer than this time period, MEG-OH could prove to be valuable coating for urinary catheters and potentially other medical devices.

Further work is necessary to determine if MEG-OH can be used successfully on urinary catheters. Firstly, actual production Foley catheters would need to be modified with MEG-OH coatings and have their surfaces characterized to determine the extent and uniformity of the MEG-OH coating. Additionally, an in vivo animal study will need to be performed to determine if there are any adverse effects from the use of MEG-OH coatings inside the body, as well as to determine if the infection rates of catheterized animals is reduced as a result of the coating. Finally, the coating will need to be put through human trials, again, in order to determine the safety and efficacy of MEG-OH coatings in reducing infection rates in patients.

## 5. Conclusions

MEG-OH surface coatings were previously found to reduce microbial (*E. coli*, *S. aureus*, *P. aeruginosa*, and *C. albicans*) fouling from a variety of species across a 1-day time period but had not been tested for periods of use longer than 3 days against any species. To remedy this, MEG-OH coatings were used on polyurethane tubing to study the reduction in bacterial fouling from Gram-negative *E. coli* and Gram-positive *S. aureus* across time periods greater than 30 days. It was found that the MEG-OH coating provided a greater than log 1.3 reduction in fouling from these bacteria across this time period, showing that MEG-OH is stable and able to continue to function for at least several weeks. It also compares very well to other types of coatings, such as PEG, SLIPS, or surface-bound enzymes across this entire time period. This clearly indicates that it would be appropriate for use on urinary catheters, as these devices are generally not implanted for longer than 1 month. Before this application can be achieved, MEG-OH-coated catheters will have to be assessed in vivo via animal models to ensure that the system does not cause any adverse reactions to mammalian tissues. Additionally, MEG-OH coatings would have to be applied to full-size urinary catheters at scale for their use in medical settings. Finally, human trials need to be conducted to ensure the safety of MEG-OH attached to implanted catheters.

## Figures and Tables

**Figure 1 biomedicines-10-00979-f001:**
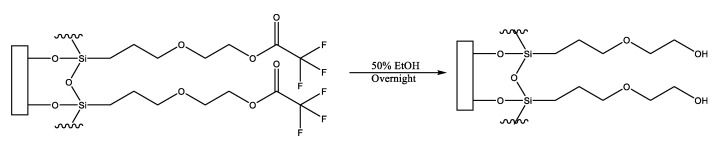
Conversion of surface bound MEG-TFA to MEG-OH.

**Figure 2 biomedicines-10-00979-f002:**
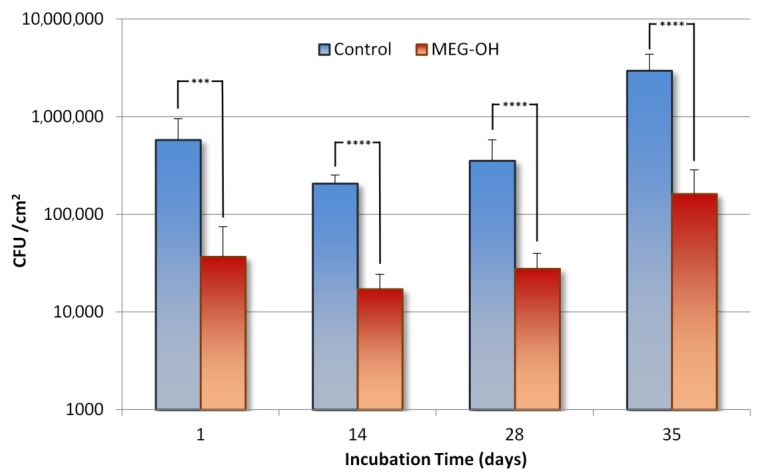
Comparison of CFU counts of *E. coli* growing on control (blue) and MEG-OH coated (red) polyurethane surfaces at days 1 (static culture), 14, 28, and 35 (under flow of 2 g/L LB medium). Asterisks represent the calculated confidence interval using a two variable *t*-test (*** = 98%, and **** = 99%+, *n* = 6).

**Figure 3 biomedicines-10-00979-f003:**
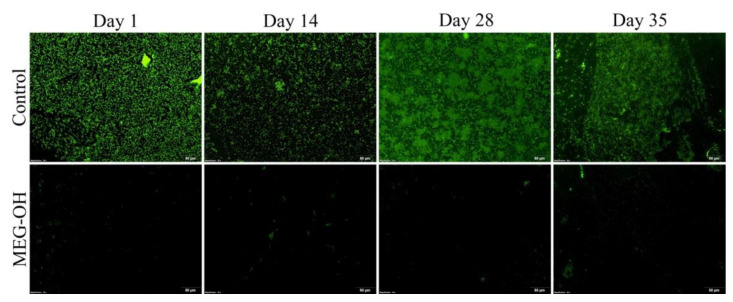
Fluorescence images of control, and MEG-OH coated surfaces that were exposed to static LB containing *E. coli* for 1 day, then at days 14, 28, and 35 under flow of 2 g/L concentration LB medium.

**Figure 4 biomedicines-10-00979-f004:**
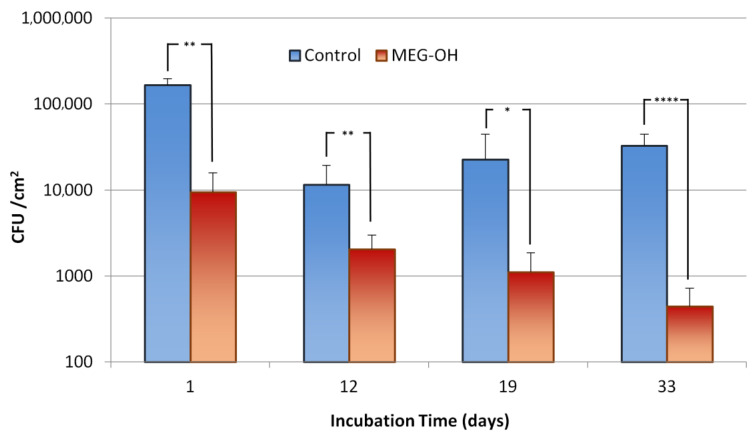
Comparison of CFU counts of *S. aureus* growing on control (blue) and MEG-OH coated (red) polyurethane surfaces at days 1 (static culture), 12, 19, and 33 (under flow of 2 g/L LB medium). Asterisks represent the calculated confidence interval using a two variable *t*-test (* = 90%, ** = 95%, and **** = 99%+, *n* = 6).

**Figure 5 biomedicines-10-00979-f005:**
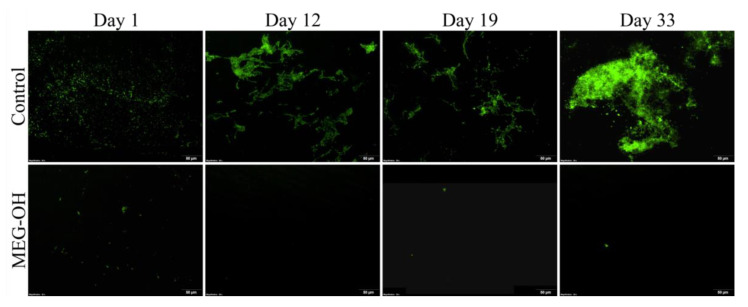
Fluorescence images of control, and MEG-OH coated surfaces that were exposed to static LB containing *S. aureus* for 1 day, then at days 12, 19, and 33 under flow of 2 g/L concentration LB medium.

## Data Availability

Not applicable.

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
