# Peer review of "Long-Term Reduction of Bacterial Adhesion on Polyurethane by an Ultra-Thin Surface Modifier"

_biomedicines, 2022, doi:10.3390/biomedicines10050979_

Round 1

Reviewer 1 Report

Dear Authors,

The work entitled  “Long-term reduction of bacterial adhesion on polyurethane by an ultra-thin surface modifier ” (biomedicines-1618127) is interesting and valuable.

The results obtained in the current study are original, interesting, and noteworthy. In my opinion, the work requires introducing the suggestions listed below:

Introduction:

  1. Lines 37-39: “There are several species that can cause CAUTIs, with Escherichia coli, Proteus mirabilis, Pseudomonas aeruginosa, Candida albicans, and Staphylococcus aureus among the most common” - this sentence must be supported by the relevant literature.
  2. Line 58: delete one of the two “for long periods”

Materials and Methods:

  1. Urinary tract infections were mentioned in the manuscript… Escherichia coli GFP (ATCC 25922GFP) does not belong to UPEC strains.  Staphylococcus aureus KR3 also is not uropathogenic. Could you explain why such strains were used in the experiments?

Results:

  1. The results have to be statistically analyzed. It would be known whether they are significant – provide relevant statistical analysis.
  2. The title of Figure 2 is unclear - my proposal of the title is: Comparison of counts of E. coli growing on the control and MEG-OH coated polyurethane surfaces in 1., 14., 28., and 35. days of the experiment.
  3. Figure 3 – change “floe” to “flow”; delete: “(top row)”, “(bottom raw)”, and concentration Figure 5 – delete: “(top row)”, “(bottom raw)”, and concentration (2 g/L); add ”static” like it is in Figure 3
  4. The title of Figure 4 is unclear - my proposal of the title is: Comparison of counts of S. aureus growing on the control and MEG-OH coated polyurethane surfaces in 1., 12., 19., and 33. days of the experiment.
  5. Figure 5 – delete: “(top row)”, “(bottom raw)”, and concentration (2 g/L); add ”static” like it is in Figure 3
  6. Line 195: should be aureus instead of S. Aureus
  7. Latin names of bacterial species should be written with italics - correct this throughout the manuscript e.g. lines 156, 165, 169, 171, 196, 248, etc.

Best regards,

Author Response

Reviewer 1:

We thank you for your kind review, and your suggestions on how to improve the paper.  We have done our best to do all the recommended changes, and have added in the requested statistical analysis.

Introduction:

Lines 37-39: “There are several species that can cause CAUTIs, with Escherichia coliProteus mirabilis, Pseudomonas aeruginosa, Candida albicans, and Staphylococcus aureus among the most common” - this sentence must be supported by the relevant literature.

  • We have added the relevant reference for this sentence.

Line 58: delete one of the two “for long periods”

  • This has been corrected.

Materials and Methods:

Urinary tract infections were mentioned in the manuscript… Escherichia coli GFP (ATCC 25922GFP) does not belong to UPEC strains.  Staphylococcus aureus KR3 also is not uropathogenic. Could you explain why such strains were used in the experiments?

  • We used these particular strains since they were approved for use with our biosafety certificate and already present in our lab, making it possible to quickly set up and perform fouling experiments. We have not added this explanation to the manuscript if that’s okay.

Results:

The results have to be statistically analyzed. It would be known whether they are significant – provide relevant statistical analysis.

  • We have performed statistical analysis on all of the CFU counts and included the calculated confidence intervals in Figures 2 and 4, as well as in the text of the results.

The title of Figure 2 is unclear - my proposal of the title is: Comparison of counts of E. coli growing on the control and MEG-OH coated polyurethane surfaces in 1., 14., 28., and 35. days of the experiment.

  • We have changed the title of Figure 2, though we did leave in the static versus flow for the days mentioned as we think that it is important information for the caption. We hope you find the new caption more understandable.

Figure 3 – change “floe” to “flow”; delete: “(top row)”, “(bottom raw)”, and concentration Figure 5 – delete: “(top row)”, “(bottom raw)”, and concentration (2 g/L); add ”static” like it is in Figure 3

  • These changes have been made.

The title of Figure 4 is unclear - my proposal of the title is: Comparison of counts of S. aureus growing on the control and MEG-OH coated polyurethane surfaces in 1., 12., 19., and 33. days of the experiment.

  • As with figure 2 we have changed the title to more closely match your proposed title.

Figure 5 – delete: “(top row)”, “(bottom raw)”, and concentration (2 g/L); add ”static” like it is in Figure 3

  • These changes have been made.

Line 195: should be aureus instead of S. Aureus

  • This has been corrected.

Latin names of bacterial species should be written with italics - correct this throughout the manuscript e.g. lines 156, 165, 169, 171, 196, 248, etc.

  • Thank you, we didn’t realize we missed so many. We have gone through and made sure to italicize all of the latin bacteria names.

Reviewer 2 Report

Dear authors,

Infections caused by indwelling devices such as urinary catheters remain still a serious problem for patients and a challenge for researchers. Submitted article by Brian De La Franier and colleagues is focused on innovation in coating of surfaces to avoid pathogens fouling. Bacterial biofilms can be formed on surfaces of medical devices and therefore to find optimal molecule for surface treatment is difficult in terms of stability, safety, and cost. Authors realized a study to detect inhibition of bacterial growth using 2-(3- 17 trichlorosilylpropyloxy)-ethyl hydroxide (MEG-OH). As they found out, reduction in colony forming units’ survival of E. coli or S. aureus was noticeable even after weeks since the beginning of the experiment with usage of MEG-OH coating. Results proved MEG-OH efficacy in reduction of biofilm formation. 
There is missing italics in names of bacteria (for example line 171, 196, 197, etc.)
Spelling mistakes: line 197 bacrtia, line 169 floe
Line 266 – In vivo - italic 
Repetition line 58: for long periods
Line 99 – I recommend using cm instead of inch 
Mainly, there is missing statistical analysis.
Authors used two different methods – CFUs and imaging, obtained results are significant, but there is at least statistic missing with number of repetition and also such kind of other molecular method to confirm surviving cells can help to support the obtained results.

Author Response

Reviewer 2:

We thank you for your kind review, and have done our best to address the mistakes you pointed out and have added in the requested statistical analysis.

There is missing italics in names of bacteria (for example line 171, 196, 197, etc.)

  • We have gone through and italicized the latin bacteria names we missed.

Spelling mistakes: line 197 bacrtia, line 169 floe

  • We have corrected these and other spelling mistakes found.

Line 266 – In vivo - italic 

  • We have italicized this.

Repetition line 58: for long periods

  • Repetition removed.

Line 99 – I recommend using cm instead of inch 

  • We have changed it from 1 inch to 2.5 cm.

Mainly, there is missing statistical analysis. Authors used two different methods – CFUs and imaging, obtained results are significant, but there is at least statistic missing with number of repetition and also such kind of other molecular method to confirm surviving cells can help to support the obtained results.

  • We have added statistical analysis for all CFU counts and included these counts in figure 2 and 4 as well as the text of the results section, and added the relevant number of samples and degrees of freedom to the methods section

Round 2

Reviewer 2 Report

Dear Authors,

Thank you for your reply and for considering my recommendation.